# Monkeypox Virus Immune Evasion and Eye Manifestation: Beyond Eyelid Implications

**DOI:** 10.3390/v15122301

**Published:** 2023-11-23

**Authors:** Francisco D. Lucena-Neto, Luiz F. M. Falcão, Adolfo S. Vieira-Junior, Evelly C. S. Moraes, Joacy P. F. David, Camilla C. Silva, Jorge R. Sousa, Maria I. S. Duarte, Pedro F. C. Vasconcelos, Juarez A. S. Quaresma

**Affiliations:** 1Department of Infectious Disease, School of Medicine, State University of Pará, Belém 66087-670, PA, Brazil; fneto640@gmail.com (F.D.L.-N.); ffalcao03@icloud.com (L.F.M.F.); adolfo.vieirat@gmail.com (A.S.V.-J.); camicost@live.com (C.C.S.); krekrodrigues@gmail.com (J.R.S.); pedro.vasconcelos@uepa.br (P.F.C.V.); 2Department of Infectious Disease, School of Medicine, Federal University of Pará, Belém 66075-110, PA, Brazil; evelly_moraes@hotmail.com (E.C.S.M.); joacydavid@yahoo.com (J.P.F.D.); 3Department of Infectious Disease, School of Medicine, São Paulo University, São Paulo 01246-904, SP, Brazil; miduarte@usp.br; 4Virology Section, Evandro Chagas Institute, Ananindeua 67030-000, PA, Brazil

**Keywords:** monkeypox virus, ocular disease, immune pathogenesis

## Abstract

Monkeypox virus (MPXV), belonging to the *Poxviridae* family and *Orthopoxvirus* genus, is closely related to the smallpox virus. Initial prodromal symptoms typically include headache, fever, and lymphadenopathy. This review aims to detail various ocular manifestations and immune evasion associated with the monkeypox viral infection and its complications, making it appropriate as a narrative review. Common external ocular manifestations of MPXV typically involve a generalized pustular rash, keratitis, discharges, and dried secretions related to conjunctival pustules, photophobia, and lacrimation. Orthopoxviruses can evade host immune responses by secreting proteins that antagonize the functions of host IFNγ, CC and CXC chemokines, IL-1β, and the complement system. One of the most important transcription factors downstream of pattern recognition receptors binding is IRF3, which controls the expression of the crucial antiviral molecules IFNα and IFNβ. We strongly recommend that ophthalmologists include MPXV as part of their differential diagnosis when they encounter similar cases presenting with ophthalmic manifestations such as conjunctivitis, blepharitis, or corneal lesions. Furthermore, because non-vaccinated individuals are more likely to exhibit these symptoms, it is recommended that healthcare administrators prioritize smallpox vaccination for at-risk groups, including very young children, pregnant women, older adults, and immunocompromised individuals, especially those in close contact with MPXV cases.

## 1. Introduction

As the world continues to recover from the waves of coronavirus (COVID-19) contagion, which led to numerous negative economic and social repercussions [1], the monkeypox (MPXV) outbreak emerges as a new threat, particularly in light of the expanding cases in non-endemic countries [2].

MPXV belongs to the *Poxviridae* family, genus *Orthopoxvirus*, and is closely related to the smallpox virus. Therefore, it is a double-stranded deoxyribonucleic acid virus that causes a zoonotic disease [3]. This virus was first registered in colonies of monkeys in 1958 in Denmark [4]. Nevertheless, only 12 years later, the first human MPXV case was documented in the Democratic Republic of Congo (DRC) [5].

In 2003, the largest certified series of zoonotic MPXV propagations occurred in the United States of America, where prairie dogs were in contact with contaminated African animals imported from Ghana. Following this contact, 71 individuals who had interactions with these pets became infected, marking it the first human MPXV incident outside of Africa [6]. Contrary to the virus’s name, monkeys are not its primary reservoir. The natural host is still unknown, but research indicates that African squirrels and rodents are likely major virus reservoirs [7].

Global efforts were effective in eradicating smallpox during the 1980s. Ever since this date, no one has manifested this illness in the world [8]. Because the smallpox vaccine provides a cross-protection of approximately 85% against monkeypox viruses, its discontinuation has contributed to the rise in MPXV cases around the globe [9,10].

MPXV has traditionally afflicted marginalized and low-income sub-Saharan African countries [11]. Since 1970, 11 African countries have reported human cases of MPXV, with a median age of 31 years [12]. In that regard, just in the DRC, 10,000 suspected and confirmed cases of this illness were reported between the years 2000 and 2009 [13]. On 7 May 2022, a critical turning point in this disease’s history occurred when the virus garnered global attention. The UK confirmed an MPXV case in an individual recently returning from Nigeria [14]. Since then, human monkeypox has become a global concern, with over 6000 confirmed cases and 22 fatalities reported across at least 87 countries (Figure 1) [15].

In addition, environmental conditions can influence zoonotic diseases. Specifically, the prevailing environmental degradation in different places worldwide, changes in the habitat of reservoir animals, expansion of agriculture, climate change, urbanization, and migration allow for more frequent contact between humans and possibly infected animals [16].

In view of recent events, the possibility of sexual transmission has become a concern in the 2022 multicountry MPXV outbreak. The first reason for sexual transmission involves unusual lesions in the genital and perianal regions [17]. Moreover, the progression of the rash does not adhere to the established pattern observed in recurrent cases in Africa [18]. Additionally, a comparison between older and more recent studies indicates a variation in genital involvement, ranging from 30% to 68% [5,19,20].

Lastly, the major clinical characteristic of monkeypox is a disseminated vesiculopustular rash. Although all ophthalmological repercussions are important, corneal scarring and loss of vision are some of the main sequelae [21]. In endemic areas, nearly 20% of MPXV-infected people manifest oedema of the eyelids or conjunctivitis [22].

## 2. Methods

We conducted a semi-systematic narrative review. We did not conduct a systematic review in accordance with the Preferred Reporting Items for Systematic Reviews and Meta-Analyses (PRISMA) guidelines because this review aims to propose and characterize several manifestations of the ocular monkeypox virus and its associated complications, making it more suitable as a narrative review. The primary databases used to retrieve salient medical literature presented in this review were PubMed, ScienceDirect, and Cochrane. The search terms, used both separately and in combination, included ‘monkeypox virus eye disease’, ‘monkeypox virus ocular manifestation’, and ‘MPXV pathophysiology’. Only articles published in English were included in this study.

## 3. Viral Agents and Reservoirs

The monkeypox virus belongs to the *Poxvridae* family and is genetically different from other members, such as smallpox and cowpox viruses [23]. It can maintain a reservoir in a wide range of hosts, a characteristic that makes it different from the smallpox virus, the variola agent [24]. Therefore, this virus can sporadically cause small and limited outbreaks in human communities, and once established, it is difficult to eradicate even via human vaccination [25,26].

Initial studies distinguished between monkeypox and variola using chorioallantoic membrane cell cultures. The smallpox virus forms a red hemorrhagic lesion on the membrane, whereas monkeypox creates a white lesion [27]. Serologically, monkeypox and variola are similar and challenging to differentiate through neutralization or hemagglutination inhibition without specific antisera to viral agents like mo and va [28]. This method identified a monkeypox virus reservoir in Central African wild primates [29].

Poxviridae family members have distinct and specific characteristics such as DNA cleavage sites, surface epitopes, and polypeptides [30]. Despite such differences, developing a rapid and reliable diagnostic test for this virus remains challenging [31].

In 2001, a genomic comparison between monkeypox and variola viruses was performed by sequencing the 197-kb genome of the monkeypox virus isolated from infected humans during an outbreak in the DRC. The central region of the genome, which encodes the essential enzymes and structural proteins of the virus, was 96.3% identical to that of the variola (smallpox) virus. In contrast, in regions encoding virulence and host-range factors, there were significant differences [32,33].

DNA maps of variola and monkeypox were so profoundly different that a spontaneous production of smallpox from monkeypox was impossible [34]. This evidence indicated that smallpox and monkeypox are not derivatives of each other, despite sharing a common ancestor [35]. It also dispelled concerns that monkeypox could mutate into smallpox, potentially compromising the long-standing success of the human vaccination program for variola [36]. The eradication of the variola virus was achieved through a successful mass vaccination strategy and the absence of animal or environmental reservoirs [37]. Although monkeypox has a broad host range, it exhibits low human transmissibility and, as shown, cannot transform into a ‘new’ variola [38,39].

Monkeypox and smallpox have similar symptoms but different fatality and human-to-human transmission rates [40]. Several factors may explain the differences in clinical presentation, epidemiology, and host selection between poxviruses. For example, the BR-203 virulence protein is encoded by monkeypox but not by smallpox and is believed to play an important role in avoiding apoptosis of infected lymphocyte cells [41]. Specifically, BR-203 is an ortholog of the myxoma virus M-T4 gene. The Central African strain of the monkeypox virus, orthologous to the BR-203 gene, encodes a full-length protein comprising 221 amino acids (aa). In contrast, the West African strain is predicted to encode only an N-terminal fragment of approximately 51 aa. When rabbits are infected with the myxoma virus, their leukocytes become infected, allowing the virus to replicate. The M-T4 protein remains in the endoplasmic reticulum (ER) and is not secreted during viral infection, partly due to a C-terminal–RDEL sequence that anchors it in the ER. Removal of this gene from the myxoma virus attenuates the virus and leads to the death of infected lymphocytes, thereby hindering the primary mode of viral spread within the host [42,43].

COP-B7R, a virulence protein encoded exclusively by monkeypox [44], resides in the ER and comprises 182 aa. The mechanism sustaining this protein remains unclear. Deletion of the B7R gene does not impact viral replication, yet a B7R-deleted vaccinia virus exhibits reduced pathogenicity in an intradermal mouse model. It is hypothesized that B7R may influence apoptosis, akin to the myxoma virus M-T4 gene, or it might interact with and stabilize in the ER of a cell-surface protein frequently involved in the immune response [44,45]. Conversely, COP-C10L, an interleukin (IL)-1β antagonist, is present in the variola virus but is fragmented in the monkeypox virus [46]. COP-C10L functions by inhibiting IL-1 receptors, thereby enabling the virus to evade IL-1 activity effects [47,48]. C10L structurally resembles the IL-1 receptor antagonist, a protein known to block IL-1 by binding to its receptors. Additionally, C10L interacts with K1L, a full-length host range protein in the monkeypox virus, indicating the significance of both proteins in host–virus interactions [48,49]. Moreover, COP-E3L, an interferon (IFN)-resistant protein, is expressed in the Central African strain of the monkeypox virus and is only a 153 aa fragment lacking the N-terminus. COP-E3L encodes a protein that has a C-terminal domain that binds double-stranded (DS) RNA and an N-terminal domain that binds Z-DNA. This domain exhibits activity impacting the IFN, a cytokine known for its antiviral properties. COP-K3L, a protein resistant to the IFN and homologous to the eukaryotic translation initiation factor 2α (eIF-2α), is absent in most monkeypox virus strains but present in The Central African strain. It is predicted to encode a fragmented protein approximately 43 aa in length, aiding the virus in inhibiting protein synthesis [41,50,51].

There are two classes of IFNs, type I IFN and type II IFN, and each type binds to its own type of receptor. Type I IFN includes IFN α and β, while type II includes IFN γ. E3L inhibits both types of IFN. Once the IFN is activated, certain signaling pathways lead to the gene expression and activation of several antiviral proteins, including protein kinase R (PKR) and 2′-5′ oligoadenylate synthetase (2′-5′ A) [50,51,52].

There is an enzyme called double-stranded RNA (dsRNA)-activated, acting like an inhibitor of translation. A DNA-dependent activator of the interferon regulatory factor phosphorylates the α subunit of eIF-2α and, therefore, inhibits initiation in the translation pathway. The function of PKR is to inhibit the step of initiation in protein translation by phosphorylating eIF-2α. Because the virus inhibits PKR and 2′-5′A, protein synthesis can continue. Because K3L is a mimic of eIF-2α, it is believed to bind the eIF-2α site and prevent the enzyme from autophosphorylating itself. If the initiation is not inhibited, protein synthesis can continue. Importantly, when the K3L gene is deleted, the virus becomes IFN-sensitive [52,53].

There are several ways in which the IFN is activated, including viral infection and exposure to dsRNA. COP-C10L, COP-E3L, and COP-K3L, present in the variola virus but fragmented in the monkeypox virus, may contribute to the observed differences in morbidity, mortality, and transmissibility between the variola and monkeypox viruses.

Three clades of MPXV are recognized: Clade I is present in the Congo Basin, causes up to 10% human mortality, and is transmitted by rodents with little human-to-human spread. Clade IIa is found in West Africa, has a low death rate, and is also a zoonosis; clade IIb is currently spreading globally through human transmission. It is unknown what genetic basis exists for these variations in virulence and transmission [54]. Poxviruses, unlike most other DNA viruses, replicate in the cytoplasm, and their large genomes usually encode 200 or more proteins with diverse functions [55].

In contrast to variola, the monkeypox virus may use many animals as reservoirs [56]. Serological studies suggest that squirrels, non-human primates, rabbits, and rats are some animal species susceptible to monkeypox infection under natural conditions [57]. However, human infection reservoirs are still unknown [58].

A study conducted in the DRC suggested that squirrels might be the primary reservoir for sustaining human infections in agricultural areas [59,60]. The seroprevalence of the monkeypox virus in these animals was up to two times higher than that in other animals surveyed, such as primates [61].

## 4. MPXV Immune Evasion

By generating viral proteins that are quickly expressed in the early phases of replication, Poxviruses suppress this host reaction [60]. These anti-apoptotic effectors exhibit varied modes of action. Specifically, they may be secreted to neutralize extracellular signals, such as the previously described antiviral cytokine, or function intracellularly to alter cell death pathway transduction [62]. For instance, the molluscum contagiosum virus strategy inhibits the initiator caspase, caspase-8 activation [63,64]. It produces two DED (death effector domain) containing proteins, MC159 and MC160, classified as viral FLICE/caspase-8 inhibitory proteins (vFLIPs). These proteins bind to FADD (Fas-associated death domain) and procaspase-8, blocking death receptor-mediated apoptotic signal transduction [65].

Orthopoxviruses have accrued an arsenal of genes that encode proteins that interfere with host cell signaling pathways that are involved in virus recognition, apoptosis, and immune regulation. Orthopoxviral proteins can antagonize pattern recognition receptors (PRRs) [66]. BCL-2-like proteins are generally conserved across orthopoxviruses—A47, B13, P1, C6, and D11 are orthologues of BCL-2-like proteins in the MPXV and prevent dsRNA from being detected by host intracellular PRRs. One of the most important transcription factors downstream of PRR binding is IRF3, which controls the expression of the crucial antiviral molecules IFNα and IFNβ. Moreover, the MPXV orthologue B16 can inhibit IFNβ signaling. Interestingly, the interferon response is known to be stronger in children and has been shown to protect against severe SARS-CoV-2 [66,67,68,69,70].

The most direct method involves the secretion of a vIFNα/βBP that binds to IFN-I prior to its interaction with the specific cellular receptor [71]. This protein, initially described in the VACV, is highly conserved across most virulent poxviruses, including the VARV, ECTV, and MPXV [70]. Comprising three immunoglobulin-like domains, its structure is distinct from host IFN-I receptors, which possess fibronectin type III domains. This distinctive structure likely endows the viral protein with potent IFN-I inhibitory capabilities, enabling it to block IFNs from various species. This contrasts sharply with the high species specificity of its cellular counterpart [70,71] (Figure 2).

Further, orthopoxviruses evade host immune responses by secreting proteins that antagonize the functions of host IFNγ, CC and CXC chemokines, IL-1β, and the complement system [72]. The poxvirus complement control protein is not just a cytokine decoy receptor; it is also a secreted immunomodulator binding C3b and C4b in solution, capable of attaching to the cell surface [71]. Like their host counterparts, viral complement inhibitors comprise three or four short consensus repeats [73]. For the VACV (termed VCP for VACV complement protein) and VARV (named SPICE for smallpox inhibitors of complement enzymes), cell surface attachment was first identified as occurring through an interaction with glycosaminoglycans (GAGs) [74].

Moreover, host complement regulatory proteins display GAG-binding activity, but the affinity for GAGs is much higher in the case of the poxvirus homologues (Figure 2) [75]. Central African strains encode the monkeypox inhibitor of complement enzyme (MOPICE) from the D14L gene [76,77]. Although MOPICE has a truncated short consensus repeat (SCR) domain, it inhibits complement activation by binding to C3 and C5 convertases [75,78]. In contrast, the West African MPXV clade does not express complement-modulating proteins [78].

Orthopoxviruses evade T cell-mediated and natural killer (NK) cell-mediated cytotoxicity. MPXV initially counters this system by secreting the orthopoxvirus Major Histocompatibility Complex (MHC) class I-like protein (OMCP), encoded by the N3R gene. OMCP mimics MHC class I molecules and binds to NKG2D [79], suppressing the usual NKG2D-dependent NK cell lysis of infected cells lacking MHC class I expression. This action consequently diminishes T-cell recognition [79,80].

MPXV encodes the MPXV viral chemokine inhibitor (vCCI), a highly abundant secreted chemokine binding protein produced in MPXV-infected cells. Its primary target is MIP-1α (CCL3). In vitro chemotaxis assays show that vCCI completely blocks rhMIP-1alpha-mediated chemotaxis and reduces the response of CD14(+) cells to rhMIP-1alpha, indicating the effective inhibition of chemokine function both in vitro and potentially in vivo [81]. The soluble vCCI, a poxvirus-encoded protein, tightly binds to human CC chemokines, impairing the host immune defense [82].

Orthopoxviruses produce a viral IL-18-binding protein (vIL-18BP) that further blocks the cytotoxic activities of NK cells. This protein binds to IL-18 and inhibits IL-18-induced IFN-γ production [83,84]. Thus, vIL-18BP inhibits a Th1 cytokine pattern necessary for the expansion of cytotoxic T lymphocytes (CTL) and NK cells, which are the major immune cell populations that effectively clear intracellular infections [85]. Furthermore, MPXV suppresses T cell-mediated immunity by triggering a state of T-cell unresponsiveness via an MHC-independent mechanism [80,81,85].

Another mechanism of orthopoxvirus immune evasion involves regulating apoptosis. Orthopoxvirus-encoded serine protease inhibitors (SPIs; serpins) have also been reported [86]. CrmA interferes with granzyme B155, which is secreted by cytotoxic T cells to initiate cell death in the virus-infected target cells; moreover, it inhibits caspase 1 and caspase 8 [87]. TNF-α (Tumor necrosis factor) is a cytokine secreted from T cells and macrophages that protects cells from viral infection and can kill those that have been infected with the virus. CrmB, a viral protein and an SPI-2 orthologue encoded by the B12R gene, functions as a decoy viral TNFR. These receptors, lacking signaling domains, are secreted and compete for the binding of TNF- α and TNFβ [88].

Moreover, the MPXV secretes proteins targeting key molecules such as IL-1ẞ, IL-1RA, IL-2R, IL-4, IL-5, IL-6, IL-8, IL-13, IL-15, IL-17, CCL2, and CCL5 [73]. Poxviruses have the ability to secrete immunomodulators that act on the cell surface [89]. vIL-18BP and vIFNα/βBP, which are viral IFN-I- binding proteins, are released from infected cells. These proteins can bind to their respective ligands in either a soluble form or while anchored to the cell surface via GAG interactions. In both scenarios, they function as cytokine decoy receptors and are associated with an upregulation of pro-inflammatory cytokines (Figure 2) [71,90].

## 5. Clinical Features Overview

The monkeypox virus produces a modulator that suppresses host T-cell responses, characterizing it as an immune modulating factor, contributing to the increased virulence and selective downregulation of host responses, specifically apoptosis, by silencing the transcription of genes involved in host immunity and associated with an upregulation of pro-inflammatory cytokines during infection [85,89]. The most pronounced effect of the poxvirus infection on the host transcriptome was observed in fibroblasts and primary human macrophages. In the fibroblasts, a significant number of key innate immune response genes were induced by polyinosinic:polycytidylic acid, poly(I:C) [91,92,93]. In uninfected cells, genes involved in alerting the innate immune system to infectious threats, including TNF-alpha, IL-1 alpha and beta, CCL5, and IL-6, were active, while they were silent in MPV-infected cells. This reflects the ability of virally encoded secreted factors to disrupt host interferon signaling [94,95]. Poxviruses are large, hindering their passage through host defenses [96]. However, due to a set of virulence gene-encoded molecules acting as modulators, the virus can infiltrate the host immune system. These proteins modulate the immune system, interfering with the cell’s ability to initiate oxidative burst and apoptotic pathways and impacting immune recognition (Figure 2) [97,98,99].

### 5.1. Systemic Presentation

Clinical manifestations occur after the incubation period, which generally ranges from 2 days to 2 weeks. The initial manifestations are characterized by prodromal features, including headache, fever, lymphadenopathy, muscle aches and backache, and respiratory symptoms such as sore throat, cough, and exhaustion (Figure 3) [100,101]. Nonspecific symptoms begin to develop one or two weeks after infection and resemble the seasonal flu. Lymphadenopathy occurs in the early stages of the illness, a distinctive hallmark differentiating human monkeypox from smallpox and chickenpox [102]. Specifically, the virus replicates at the inoculation site and then spreads to regional lymph nodes. Following a period of initial viremia, the virus spreads to other body organs. Notably, the MPV has a similar morphology to other known orthopoxviruses and generally does not show many mutations compared to RNA viruses like HIV (human immunodeficiency virus) or SARS-CoV-2 [103,104].

Lymphadenopathy is due to the initial activation of the immune system and, as well as lymphoid depletion, results from a monkeypox infection due to changes in the number of lymphocytes, including NK cells [105]. The acquired immune system in poxvirus infection down-regulates the expression of the class I MHC receptors. For instance, the molluscum contagiosum virus encodes an MHC I homolog (MC080R), resulting in ER-retention of the host MHC-I and thereby a reduced cell surface presentation, potentially acting as a ligand for inhibition of the NK cell receptors, serving as an antagonist of the activating receptor NKG2D. Finally, the homolog class I MHC receptors are endogenous viral antigens that evade the virus from circulating CD8 CTLs [91,98].

The differential diagnosis for monkeypox includes secondary syphilis, due to the involvement of palms and soles—along with varicella, herpes, chancroid, scabies, hand-foot-mouth disease, and medication-associated allergies [106,107]. Monkeypox is generally self-limited, with symptoms typically persisting for 2–4 weeks [108]. Clinically, severe cervical, postauricular, submandibular, and inguinal lymphadenopathy distinguish monkeypox from smallpox [109].

The infection begins with a prodromal stage, leading to a distinctive eruption. Typically, flu-like symptoms are followed by a fever that declines within the first day or up to three days. Subsequently, a rash appears 1–3 days later, initially on the face and extremities, and then spreads in a centrifugal pattern across the body [110]. Skin lesions progress through stages of macules, papules, pustules, or vesicles [111], resembling the progression seen in smallpox infections. The dermatitis is monomorphic, with lesions typically evolving from macular to papular, then to vesicular, and eventually to pustular. Most cases present as vesiculopustular monomorphic skin eruptions less than 2 cm in size, varying from a few to thousands [19].

The lesions are indistinguishable from those of ordinary smallpox. Microscopically, they exhibit all characteristics of a variola virus infection, including epidermal hyperplasia, ballooning degeneration of keratinocytes with intracytoplasmic inclusions, intraepidermal vesicles and pustules, and crust formation. Dermal changes, similar to smallpox, feature oedema and infiltration with lymphocytes, macrophages, and, to a lesser extent, neutrophils and eosinophils [20,112].

The lesions are primarily triggered by a T-cell response dominated by CD4+ T-helper cells, probably directed against the infected keratinocytes [112]. The skin lesions develop with a time delay because T-cell priming in the lymph node is probably required first. The subsequent cytotoxic CD8+ T-cell response results in the ballooning degeneration and necrosis of the infected keratinocytes, a known trigger for the recruitment of neutrophils, which dominate in the lesions as they progress [113,114]. Furthermore, the extensive diapedesis of neutrophilic granulocytes from the postcapillary venules likely results in the occlusion of superficial vessels, exacerbating the ulcerations observed in the later stages. This phenomenon, known as immunothrombosis, has been described in other viral diseases, such as SARS-CoV-2 [113,114,115].

Lesions in the oral cavity cause difficulties in eating and drinking and, thus, decrease the nutritional intake [116]. In a week, the skin lesions commonly become crusts, and when they have fallen off, the skin is left with pitted scars, hyperpigmentation, or hypopigmentation [117]. All manifestations are followed by pain, which is more severe than the other clinical findings [118]. Furthermore, oedema is prominent at the margins of necrotic areas. Eventually, inflammation and necrosis of the superficial dermis predominate, and the destruction of the sebaceous glands and follicles is evident [119]. The disruption of the skin due to the encrustation of these lesions is a matter of concern as it may cause bacterial superinfections, including encephalitis and pneumonitis [120].

Respiratory symptoms include prodromal features, such as nasal congestion, cough, and sore throat [121]. Some patients present with pulmonary distress or bronchopneumonia in the late course of the illness, suggesting a secondary infection of the lungs. Importantly, the aggravated immunologic reactions may result in septic shock and life-threatening medical conditions [122]. Additionally, gastrointestinal symptoms occur by the second week of the illness and are characterized by vomiting and diarrhea, which can contribute to severe dehydration [123]. The mouth and throat ulcers cause difficulties with nutrition [124].

Lesions in the anogenital area can be accompanied by anal proctitis, tenesmus, and pain during defecation. Most patients with anogenital involvement present genital lesions, including penile rash and groin lesions [125]. These lesions primarily consist of genital ulcers and secondary bacterial infections [126]. In cases of HIV co-infection with immunosuppression, the monkeypox disease’s natural history alters, often resulting in a more severe course [127].

### 5.2. Ocular Manifestation and Therapy

The monkeypox disease often presents with various ophthalmic manifestations, including common symptoms like frontal headaches affecting the orbits [128]. Lesions typically involve the periorbital and orbital skin, accompanied by conjunctivitis and eyelid oedema [129]. Conjunctivitis, affecting mainly men and children under 10, is the most frequent ophthalmological symptom and may indicate the illness’s progression due to its association with corneal scarring, potentially leading to blindness [130,131]. Approximately 20% of patients experience conjunctivitis, causing significant but temporary distress, often accompanied by nausea, chills/sweating, mouth ulcers, sore throat, lymphadenopathy, and fatigue [132,133].

Jezek Z et al. showed that conjunctivitis was more common among patients affected by animal MPXV (20.3%) than in those affected by human MPXV (16.4%). Meanwhile, lesions on the conjunctiva and on the margins of the eyelids were seen with a greater incidence among unvaccinated patients with confirmed MPXV (nearly 25%) [134]. Additionally, conjunctivitis appears to be predictive of the illness course. Indeed, in one study, about 50% of patients with conjunctivitis were reported as ‘bed-ridden’, in contrast to nearly 15% of patients without reported conjunctivitis [131].

External manifestations are more common in eye lesions during a monkeypox infection. The eyelids may be severely affected, sometimes unable to open for a few days due to a generalized pustular rash, discharges, and dried secretions [135]. Conjunctival pustules, often accompanied by photophobia, lacrimation, and pain, are also present [136]. Obliterating vascular changes typically cause intense ischemic pain, likely correlating with the sometimes severe pain reported by patients. Hence, it is plausible that both direct viral cytopathic effects and the host’s immunological response contribute equally to the clinical and histological stage-like presentation of MPXV [137]. Conjunctival phlyctenules, stemming from a type IV hypersensitivity reaction (cell-mediated), are occasionally reported (Figure 3) [138,139].

However, it is crucial to note that MPXV infection typically manifests with clinical symptoms of an external disease, originating from virus replication at the inoculation site, and subsequently spreads to regional lymph nodes, compromising the protective barrier of the skin and mucosal surfaces. This often affects the eye and eyelids, contrasting with viruses such as the herpes virus, SARS-CoV-2, and dengue, where clinical characteristics include both an external disease and intraocular conditions like uveitis and retinitis.

Additionally, various ophthalmic manifestations such as iritis/iridocyclitis, retinitis/chorioretinitis, optic neuritis, ophthalmoplegia, and dacryocystitis have been observed in smallpox (variola) infection and following smallpox vaccination with vaccinia. However, these symptoms have not been reported in the MPXV, and it is currently unknown whether they can occur in human monkeypox infections [140].

Ocular complications in relation to the current outbreak have been extremely rare (<1%). Cases reported from 27 EU/EEA nations showed incidences of MPXV ocular complications ranging from 9 to 23%, higher than in endemic countries, with over 20,000 cases as of 27 September 2022. Similar incidences were observed in other studies during the current outbreak: 2 out of 197 cases in London (1%), 2 out of 185 in Spain (1.1%), and 2 out of 264 in France (0.8%) [141]. Of 528 subjects, 3 had conjunctival lesions, with two requiring hospitalization. A French case series of 264 participants also found ocular involvement in 2 hospitalized patients, highlighting it as a severe manifestation of the illness [142].

These retrospective analyses may underestimate the prevalence of ocular MPXV. During the 2010–2013 clade I outbreak in the DRC, 23.1% of cases reported conjunctivitis, which were more frequently observed in patients younger than 10 years [143].

Entry receptors for MPXV have not been clearly identified, although it was suggested that viral entry is dependent on the viral strain and host cell type and involves multiple surface receptors, including chondroitin sulphate or heparan sulphate [144]. The pathogenesis and mechanism of action of the MPXV are similar to those of the variola virus (VARV) and vaccinia virus (VACV). The MPXV, like other poxviruses, likely infects a broad range of mammalian cells without requiring specific host receptors. In the VACV, surface proteins H3, A27, and D8 are linked to viral binding. Both EEV (extracellular enveloped virus) and IMV virions penetrate the host membrane through binding and micropinocytosis [144,145].

MPXV virions utilize glycosaminoglycans as host receptors (Figure 4). Three proteins identified as viral entry facilitators may aid MPXV entry into host cells via receptor binding and membrane fusion [110]. The first, protein L1, a virus membrane protein, likely binds to host-cell entry receptors. This envelope protein attaches to the cell surface through the entry/fusion complex (EFC) and is essential for merging the virus and host-cell membranes during viral entry [146]. Another MPXV cell surface-binding protein, E8L, is believed to bind to host-cell surface chondroitin sulphate proteoglycans (CSPGs), mediating the adsorption of intracellular mature virus (IMV) virions to cells [147]. The MPXV envelope protein H3L, studied in vitro and in vivo on the VARV, plays crucial roles in virus adsorption to cell surface heparan sulphate and IMV morphogenesis [144].

Glycosaminoglycans and proteoglycans have been shown to participate and/or regulate various signaling pathways, such as the TGF-β, JNK/p38, FAK, and ERK pathways. TGF-β, a family of dimeric proteins, comprises three isoforms—TGF-β1, TGF-β2, and TGF-β3—expressed by corneal and limbal epithelia, conjunctiva, and stromal keratocytes. It plays a crucial role in ocular surface homeostasis and pathology by inducing the synthesis and secretion of components, promoting adhesion complex formation in corneal epithelial cells, increasing protease inhibitor expression, and recruiting monocytes/neutrophils during inflammation [148,149].

The smallpox virus is actively triggered in tears during conjunctivitis and expresses an epidermal growth factor and host range genes that play a role in pathogenesis and cell tropism, a mechanism similar to the MCV [150,151]. Furthermore, the poxvirus double-stranded DNA produces proteins that inhibit human antiviral immunity (MC132; p28 Ubiquitin ligase, Histone deacetylase protein), thus preventing the development of the innate immune response and contributing to the persistence of skin lesions [152,153,154].

An acute viral infection leads to characteristic changes in ocular target tissues. Infection of the eyelid skin results in vesicle and ulcer formation [155]. A conjunctival viral infection causes a serous discharge, hyperplasia of conjunctival lymphoid follicles, and enlargement of the corresponding draining lymph nodes [156]. A viral infection in the corneal epithelium leads to punctate epithelial cytopathic changes, observable with 1% sodium fluorescein eye drops biomicroscopically, presenting as swollen infected epithelial cells (punctate epithelial keratitis) and the loss of individual epithelial cells (punctate epithelial erosions) [157].

The characteristic rash of monkeypox often involves the peri-orbital and orbital skin. A study analyzing 282 monkeypox patients showed frequent but temporary oedema of the eyelids and conjunctivitis [151]. Among these, 17% of unvaccinated and 13% of smallpox-vaccinated patients had focal lesions on the conjunctiva and eyelid margin. In contrast, 30% of non-vaccinated versus 7% of vaccinated individuals experienced blepharitis [134].

Photophobia was reported for more than 20% of affected patients [20]. Jezek et al. (1988) showed unilateral or bilateral blindness, and weak vision was observed in 10% of primary (patients infected from an animal source) and 5% of secondary cases (between one week and three weeks days after exposure, which may have occurred due to person-to-person transmission) [134].

A virus, such as the variola virus, can replicate successfully in the corneal epithelium and stromal keratocytes, potentially causing ulcerative keratitis and perforation. Similarly, the monkeypox virus, akin to variola and vaccinia viruses, can inoculate and replicate within the corneal tissue. In addition to common presentations of the MPXV-related ophthalmic disease, providers should be aware of the potential for restrictive periorbital gangrene, corneal perforation, and globe compromise, particularly in immunocompromised patients, which may exacerbate the MPXV disease state. US CDC (Centers for Disease Control and Prevention) data indicate that MPXV patients co-infected with HIV are more likely to require hospital-level care (8% of patients) compared to immunocompetent individuals (3% of patients). Patients with HIV and CD4+ counts lower than 200 cells per μL are especially at high risk for severe disease [158,159].

Monkeypox is typically a self-limiting disease, with benefits noted from simple therapies for ocular complications [21,160]. In summary, anterior segment eye disease due to the monkeypox virus may respond effectively to trifluridine 1% eye drops, topical antibiotic prophylaxis, ocular lubricants, and systemic tecovirimat, as per US CDC guidelines. To reduce the autoinoculation risk on the ocular surface, MPXV patients should maintain hand hygiene, refrain from touching their eyes, and avoid contact lens use [161].

We should highlight the importance of the vaccine by increasing public awareness, as the virus can cause severe conditions. Moreover, it is important to remember that complications of monkeypox occur more frequently among unvaccinated people [162]. Following a confirmed diagnosis of MPXV, ophthalmologists should be alert to any new ocular symptoms. Enhancing the accessibility of ophthalmologic resources in MPXV-endemic regions may reduce the potential for affected patients to experience serious visual sequelae [22].

We strongly recommend that ophthalmologists include MPXV as a part of their differential diagnosis when they encounter similar cases presenting with ophthalmic manifestations such as conjunctivitis, blepharitis, keratitis, or corneal lesions, because the majority of MPXV-associated ophthalmic manifestations surpass the rare event assumption (5%) and given the continuous and rapid increase in the number of cases [22,163]. Additionally, as non-vaccinated individuals have a greater tendency to exhibit these symptoms, we urge healthcare administrators to rethink the use of the smallpox vaccination for at-risk groups [22,164].

## 6. Conclusions

Members of the Poxviridae family display unique viral characteristics, including DNA cleavage sites, surface epitopes, and polypeptides. The core region of the MPXV genome, coding for essential viral structural and enzymatic proteins, shares 96.3% of its sequence with the smallpox virus, the prototype species of Poxviridae. By the 1980s, a worldwide effort had successfully eradicated the wild-type smallpox virus. The smallpox vaccine provides approximately 85% cross-protection against monkeypox viruses, and the cessation of smallpox vaccine administration contributes to the global increase in MPXV cases.

MPXV infection begins with a prodromal stage, followed by a characteristic eruption. The dermatitis lesions typically start as macular, then progress to papular, and eventually become vesicular and pustular. These may affect the anogenital area and include lesions in the oral cavity, causing pain and difficulties in eating. Additionally, respiratory and gastrointestinal symptoms, often leading to more severe clinical findings, may occur, particularly in individuals not vaccinated against smallpox.

The double-stranded DNA of the poxvirus produces proteins that inhibit human antiviral immunity, thereby hindering the development of an innate immune response and contributing to the persistence of skin lesions. Acute viral infection induces characteristic changes in ocular target tissues. Monkeypox and smallpox exhibit similar symptoms, but monkeypox has lower fatality and human-to-human transmission rates. Several factors might account for the differences in clinical presentation, epidemiology, and host selection among poxviruses. For example, BR-203 and COP-B7R are both virulence proteins; COP-C10L and COP-3EL are IL-1β antagonists; and COP-3KL is an IFN-resistance protein.

One of the key transcription factors activated by PRR binding is IRF3, which regulates the expression of the essential antiviral molecules IFNα and IFNβ. The MPXV orthologue B16 has been shown to inhibit IFNβ signaling. Additionally, orthopoxviruses counteract host immune responses by secreting proteins that interfere with host IFNγ, CC and CXC chemokines, IL-1β, and the complement system. They also evade T cell-mediated and NK cell-mediated cytotoxicity, suppressing NKG2D-dependent NK cell lysis of infected cells lacking MHC class I expression, thereby diminishing T-cell recognition. MPXV encodes a secreted chemokine binding protein known as the MPXV viral chemokine inhibitor (vCCI). A prime target of the MPXV vCCI is MIP-1α (also known as CCL3), which is produced in abundance and secreted by MPXV-infected cells.

Moreover, the MPXV secretes proteins that target key inflammatory molecules, including IL-1ẞ, IL-1RA, IL-2R, IL-4, IL-5, IL-6, IL-8, IL-13, IL-15, IL-17, CCL2, and CCL5. Poxviruses can also release immunomodulators functioning on the cell surface. The vIL-18BP and vIFNα/βBP are secreted from infected cells and bind to their respective ligands in either a soluble form or anchored to the cell surface through GAG interactions, acting as cytokine decoy receptors in both cases.

When encountering cases presenting with ophthalmic signs, including conjunctivitis, blepharitis, keratitis, or corneal lesions, ophthalmologists are advised to consider MPXV as part of their differential diagnosis. MPXV infection typically presents with symptoms of an external disease related to virus replication at the inoculation site, spreading to regional lymph nodes and disrupting the protective barrier of skin and mucosal surfaces, often affecting the eyes and eyelids. The monkeypox disease can lead to various ophthalmic manifestations, particularly frontal headaches involving the orbits. Conjunctivitis is the most common ophthalmological finding, especially in patients with photophobia, either alone or in combination with other symptoms. Corneal infections may cause severe keratitis, scarring, and potentially permanent vision loss. In some cases, unilateral or bilateral blindness and diminished vision have been observed.

The monkeypox virus seems capable of local inoculation and replication within the corneal tissue. In addition to the more common presentations of MPXV-related ophthalmic disease, providers should be aware of the potential for restrictive periorbital gangrene, corneal perforation, and globe compromise, particularly in immunocompromised patients. This underscores why we urge healthcare administrators to redistribute smallpox vaccination for at-risk groups, including very young children, pregnant women, the elderly, and immunocompromised individuals among close contacts of MPX cases. It is vital to enhance the public awareness about the vaccine’s importance, as the virus can lead to severe conditions.

In summary, anterior segment eye disease secondary to the monkeypox virus may respond effectively to trifluridine 1% eye drops, topical antibiotic prophylaxis, ocular lubricants, and systemic tecovirimat, as per US CDC guidelines. To minimize the risk of autoinoculation of the ocular surface, patients with the MPXV should maintain hand hygiene, refrain from touching their eyes, and avoid using contact lenses.

## Figures and Tables

**Figure 1 viruses-15-02301-f001:**
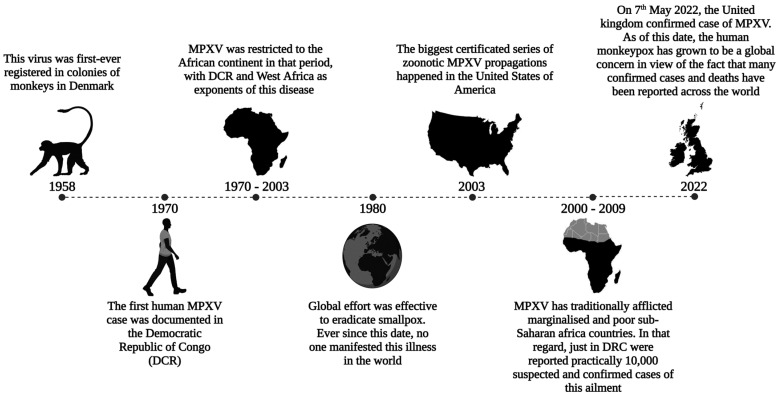
Brief history of MPXV dissemination: The global effort effectively eradicated smallpox by the 1980s. However, discontinuing the smallpox vaccine has contributed to the rise in MPXV cases worldwide, indicating a Poxviridae outbreak. Sporadic cases linked to travel or imported animals were reported prior to 2022. MPXV: monkeypox virus; DCR: Democratic Republic of Congo.

**Figure 2 viruses-15-02301-f002:**
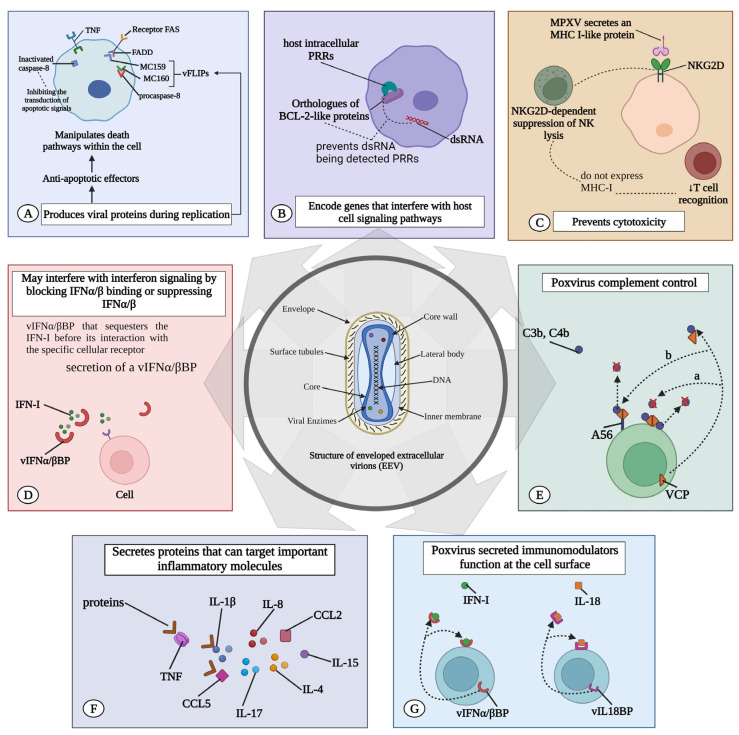
MPXV immune evasion. Immune evasion of the MPXV can occur by several mechanisms: (**A**) MPXV induces the expression of viral proteins that suppress the host reaction. These antiapoptotic effectors can be secreted and neutralize signals emanating from the extracellular environment, or they can act to manipulate the transduction of cell death pathways within the cell. (**B**) They encode proteins that can antagonize pattern recognition receptors (PRRs). BCL-2-like proteins in MPXV prevent double-stranded RNA (dsRNA) from being detected by host intracellular PRRs. (**C**) Prevent T cell-mediated and natural killer (NK) cell-mediated cytotoxicity. MPXV secretes MHC class I-like protein (MHC-I) that binds to NKG2D, suppressing typical NKG2D-dependent NK lysis of infected cells that do not express MHC-I, thereby reducing T-cell recognition. (**D**) May interfere with interferon signaling by blocking IFNα/β binding or suppressing IFNα/β. vIFNα/βBP that sequesters the IFN-I before its interaction with the specific cellular receptor. (**E**) The poxvirus complement control protein is not just a cytokine decoy receptor; it is also a secreted immunomodulator that binds to C3b and C4b in solution and can attach to the cell surface. (**F**) MPXV secretes proteins targeting key molecules such as IL-1β, IL-1RA, IL-2R, IL-4, IL-5, IL-6, IL-8, IL-13, IL-15, IL-17, CCL2, and CCL5. (**G**) It produces an IL-18 binding protein (vIL18BP) that further blocks the cytotoxic activities of NK. vIL18BP inhibits IL-18-induced interferon (IFN)-γ production and a pattern of Th1 cytokines required for the expansion of cytotoxic T lymphocytes (CTL) and NK cells. TNF: tumor necrosis factor; MPXV: monkeypox virus; IFN: interferon; IL: interleukin.

**Figure 3 viruses-15-02301-f003:**
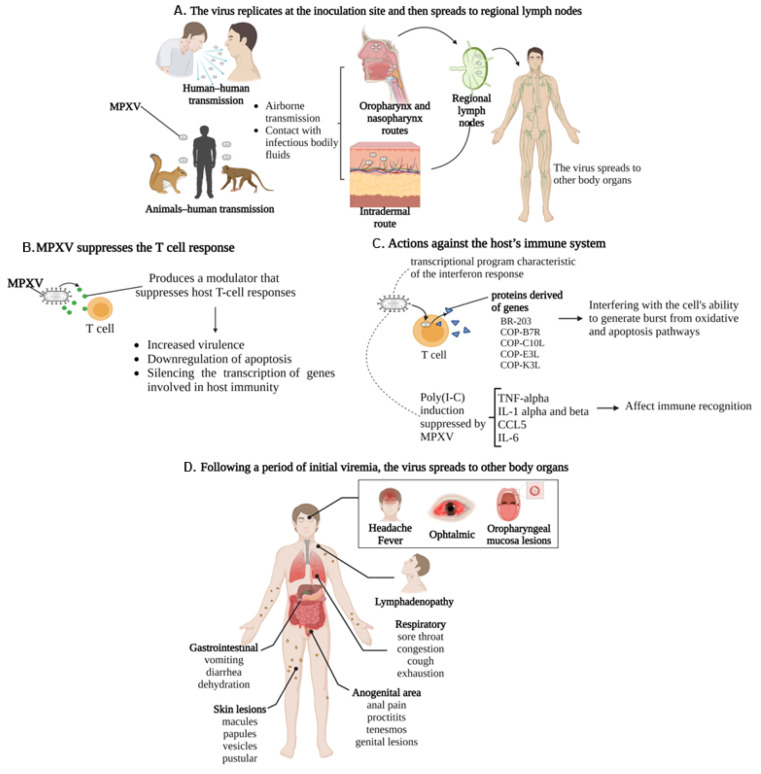
Pathophysiology and clinical presentation of MPXV: (**A**) MPXV transmission can occur through contact with infectious bodily fluids (human-to-human transmission) or animal–human transmission. Serological studies suggest that squirrels, non-human primates, rabbits, and rats are some animal species susceptible to infection. The virus replicates at the site of inoculation and then spreads to regional lymph nodes and subsequently to other organs in the body. (**B**) MPXV suppresses the host T-cell response by increasing virulence, downregulating apoptosis, and silencing the transcription of genes involved in host immunity. (**C**) MPV potently induced a transcriptional program characteristic of the interferon response, generating proteins derived of genes BR-203, COP-B7R, COP-C10L, COP-E3L, and COP-K3L, interfering with the cell’s ability to generate from oxidative burst and apoptotic pathways—Poly(I:C) induction of genes involved in alerting the innate immune system to the infectious threat, including TNF-alpha, IL-1 alpha and beta, CCL5, and IL-6, were suppressed by infection with live MPV. (**D**) Inducing the classic systemic presentation.

**Figure 4 viruses-15-02301-f004:**
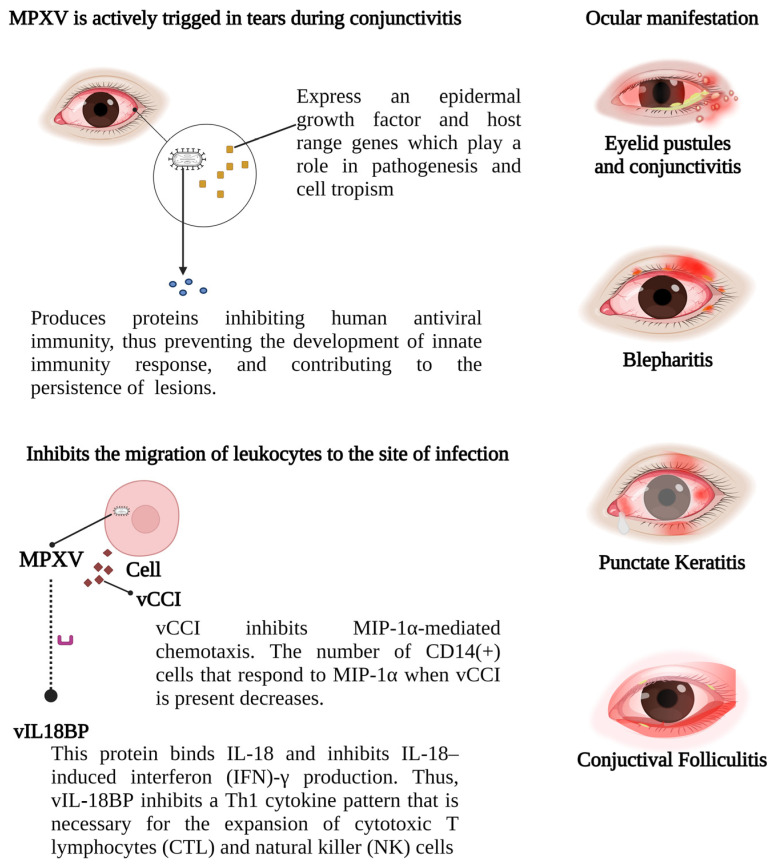
MPXV ocular manifestations: A proposed mechanism for MPXV pathogenesis, similar to MCV, involves the production of CC (two adjacent cysteine residues) chemokine, resembling monocyte inhibitory protein (MIP)-1α/β. This chemokine antagonizes MIP-1α chemotaxis and inhibits chemotaxis by various CXC (amino acid between two cysteine residues) chemokines and other CC chemokines, thereby hindering leukocyte migration to the infection site. Acute viral infection triggers typical changes in ocular target tissues. Eyelid skin infection leads to vesicle and ulcer formation, while conjunctival infection causes serous discharge, conjunctival lymphoid follicle hyperplasia, and enlargement of associated lymph nodes. MPXV: monkeypox virus; IFN: interferon; IL: interleukin; vCCI: viral chemokine inhibitor; MIP-1a: macrophage inflammatory protein-1 alpha; vIL18BP: viral interleukin-18 binding protein; NK: natural killer; CTL: cytotoxic T lymphocyte.

## Data Availability

The data supporting the conclusions of this research manuscript are all present within the article.

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
