# Peer review of "Monkeypox Virus Immune Evasion and Eye Manifestation: Beyond Eyelid Implications"

_viruses, 2023, doi:10.3390/v15122301_

Round 1

Reviewer 1 Report

Comments and Suggestions for Authors

The authors have reviewed the ocular manifestations associated with monkeypox virus infection. However, the topic is not new; several recent articles are available that evaluate this topic. The authors need to clearly emphasize what new insights they bring.

The manuscript is well-written and organized. The bibliography includes recent and relevant articles related to the discussed theme. The authors have also included figures to complement the article. The figures are well-executed, with the note that the resolution of Figure 1 should be modified as it is more challenging to read. A correct resolution is needed for all figures. Please clarify „the source” for the figures (are new? are original? are addapted?)

The Conclusions section is too lengthy and needs to be restructured. This section should encompass clear and concise conclusions regarding the review.

Comments on the Quality of English Language

Please check again for all needed corrections. If possible, do ask an English native colleague to read the manuscript.

Reviewer 2 Report

Comments and Suggestions for Authors

The review entitled “Eye manifestations of monkeypox virus: beyond eyelid implications” is very easy and nice to read and presents an overview on the clinical presentation of the monkeypox virus infection. However, the reported data in this review are not a novelty in this topic and it appears as a short history of monkeypox infections and its related clinical presentations.

It is not focused enough on eye manifestations of the monkey pox virus albeit this topic was announced in the title, and more information about updated epidemiology focused on eye involvements during monkeypox associated infection should be included.

More detailed data on the number of subjects infected exhibiting ocular lesions, vaccinated and infected subjects with eye involvement, immunocompromised infected subjects with eye lesions, etc.

Clinical difference, if any, in the eye involvement during other poxvirus infections should be discussed. A deeper analysis about the pathogenesis and related molecular mechanisms of eye damage should be reported.

Moreover, additional data on the geographic areas where eye manifestation were reported and comments on the difference, if any, in such reports about the specific topic of eye involvement during monkeypox infection.

The use of local therapy for the ocular lesions should be described together with data on the response to the drugs should be included even if the drugs are not antiviral specific molecules.

Comments on the Quality of English Language

Minor English typing errors should be corrected.

Reviewer 3 Report

Comments and Suggestions for Authors

This is a very interesting manuscript; however, it requires a few major modifications. There is quite a bit of discussion of ocular features of monkeypox viral infection with very little discussion of the pathogenesis of the MPXV ocular infection. There is no discussion of the target cells and their receptors in the cornea and conjunctiva that are utilized by MPXV to attach and gain entry to produce a productive infection. The authors should provide a detailed discussion of the pathogenesis of ocular MPXV infection starting with the role of viral surface binding proteins such as L1, E8L, and H3L in the attachment and subsequent entry of MPXV into permissive target cells, as well as discuss the tropism of MPXV for epithelial cells and fibroblasts in the infected eye. A discussion regarding evasion of the host immune defense is incomplete without discussing the role of Orthopoxvirus major histocompatibility complex class I-like protein (OMCP) and MPXV-encoded cytokine response-modifying protein B (CrmB) in facilitating evasion of the antiviral immune defenses.

Another concern I have with this manuscript is that the authors have not, in some instances, given appropriate credit in citing the literature. The authors should verify the accuracy of a few of the sentences with errors in the cited reference and provide an appropriate citation. The authors should check that all citations relate to and support the claims made. 

Page 1, line 18: Consider rendering “several manifestations of the ocular monkeypox virus” as “several ocular manifestations of monkeypox viral infection?

Page 2, line 28: The authors should elaborate on these “at-risk groups”

Page 2, lines 53 (see figure 1 and line 65): Consider rendering “poor” as “low income”

Page 2, lines 64 - 68: The authors should reword this sentence to render it more comprehensible.

On page 3, the authors highlighted a good number of viral proteins without providing adequate discussion of their respective roles in viral pathogenesis. What is the function of COP-B7R? Why is COP-E3L and its role in evading the antiviral immunity via inhibition of type I and II interferon not highlighted in this manuscript?

On page 4, the authors stated that viral proteins are interfering with oxidative and apoptotic pathways in figure 2. What are these viral proteins produced during MPXV infectious process?

Page 6, line 226: Which proteins are you referring to?

Page 6, line 234: isolated swollen epithelial cells or infected swollen epithelial cells?

Page 6, line 237: The author stated, “Recent studies of this zoonotic disease” and only one reference is cited. The authors should cite two or more references for the “recent studies”.

On page 7, the authors focused heavily on discussing MPXV\-induced immune evasion strategies. There is no discussion of the cellular receptors on permissive target cells that could promote productive infection of target cells in the cornea and conjunctiva.

Page 7, line 262: Render “CCL4” as “CCL3”. This should apply to line 369.

Page 7, lines 287 – 289: Infection has been shown to be associated with an upregulation of pro-inflammatory cytokines.

Page 7, lines 289 – 293: The authors should check that this citation relates to and support the claims made about viral IFN-I-binding protein.

On page 8, SPICE is discussed but there is no discussion of MOSPICE, a viral protein present in the Central African strain of MPXV. This viral protein inhibits C3 and C5 convertases. Thus, MOSPICE inhibits the generation of the alternative and classical pathways of complement.

Page 8, line 300: Cite the original source.

Page 8, figure 4: What are these secreted proteins?

 Page 9, line 369: Render “CCL4” as “CCL3”.

Comments on the Quality of English Language

A few sentences need to be reworded in order to render them more comprehensible.

Round 2

Reviewer 1 Report

Comments and Suggestions for Authors

As the issues were addressed - the manuscript is publishable.

Reviewer 2 Report

Comments and Suggestions for Authors

In the revised manuscript all the reviewers' comments were addressed. Some typing errors still reman in the  English language.

Comments on the Quality of English Language

Some typing errors are still present in the English language.

Reviewer 3 Report

Comments and Suggestions for Authors

This is a very interesting manuscript; however, the authors have not addressed my concerns sufficiently to make this manuscript suitable for publication.

MAJOR ISSUES

One of my concerns with this manuscript is that the authors have not, in some instances, given appropriate credit in citing the literature. The authors should verify the accuracy of a few of the sentences with errors in the cited reference and provide an appropriate citation. The authors should check that all citations relate to and support the claims made. A few sentences need to be reworded in order to render them more comprehensible.

Page 3, lines 83 – 85: In addition to reference #5, the authors should cite the following sources:

1. Dimie Ogoina, Michael Iroezindu, Hendris Izibewule James, Regina Oladokun, Adesola Yinka-Ogunleye, Paul Wakama, Bolaji Otike-odibi, Liman Muhammed Usman, Emmanuel Obazee, Olusola Aruna, Chikwe Ihekweazu, Clinical Course and Outcome of Human Monkeypox in Nigeria, Clinical Infectious Diseases, Volume 71, Issue 8, 15 October 2020, Pages e210–e21

2.     Yinka-Ogunleye A, Aruna O, Dalhat M, Ogoina D, McCollum A, Disu Y, Mamadu I, Akinpelu A, Ahmad A, Burga J, Ndoreraho A. Outbreak of human monkeypox in Nigeria in 2017–18: a clinical and epidemiological report. The Lancet Infectious Diseases. 2019 Aug 1;19(8):872-9.

Page 4, line 166: Which strain of monkeypox virus is predicted to express COP-K3L?

Page 5, line 222: Render “(vIFNa/BBP)” as “(vIFNα/βBP)”

Page 6, line 275: Render “TNF” as “TNF-alpha”

Page 6, line 279: Render “(vIFNa/BBP)” as “(vIFNα/βBP)”

Page 6, line 276: Render “inflammatory molecules” as “molecules”, since IL-1RA is a natural anti-inflammatory molecule that neutralizes the pro-inflammatory activity of IL-1.

Page 6, lines 278 – 282: The cited reference does not support the claim made in this sentence. The authors should cite the following references:

1.     Xiang Y, Moss B. Molluscum contagiosum virus interleukin-18 (IL-18) binding protein is secreted as a full-length form that binds cell surface glycosaminoglycans through the C-terminal tail and a furin-cleaved form with only the IL-18 binding domain. J Virol. 2003 Feb;77(4):2623-30. 

2.     Hernaez B, Alcami A. New insights into the immunomodulatory properties of poxvirus cytokine decoy receptors at the cell surface. F1000Res. 2018 Jun 11;7:F1000 Faculty Rev-719.

Page 6, line 276: Because IL-1RA is a neutralizer of the pro-inflammatory activity of IL_1, please render “inflammatory molecules” as “molecules” This should apply to page 7 line 300 and page 14 line 596.

Page 6, line 296: Render “(vIFNa/BBP)” as “(vIFNα/βBP)”

Page 7, lines 287 – 289: Infection has been shown to be associated with an upregulation of pro-inflammatory cytokines.

Page 7, lines 289 – 293: The authors should check that this citation relates to and support the claims made about viral IFN-I-binding protein.

Page 7- 8, lines 307 – 310: The cited reference does not support the claim made in this sentence. Please provide the appropriate source for this sentence. Check if the following references including reference #98 are a good fit:

1.     Harte MT, Haga IR, Maloney G, Gray P, Reading PC, Bartlett NW, Smith GL, Bowie A, O'Neill LA. The poxvirus protein A52R targets Toll-like receptor signaling complexes to suppress host defense. J Exp Med. 2003 Feb 3;197(3):343-51.

2.     Stack J, Haga IR, Schröder M, Bartlett NW, Maloney G, Reading PC, Fitzgerald KA, Smith GL, Bowie AG. Vaccinia virus protein A46R targets multiple Toll-like-interleukin-1 receptor adaptors and contributes to virulence. J Exp Med. 2005 Mar 21;201(6):1007-18.

3.     Yu H, Bruneau RC, Brennan G, Rothenburg S. Battle Royale: Innate Recognition of Poxviruses and Viral Immune Evasion. Biomedicines. 2021 Jul 1;9(7):765. (This reference #98)

Page 11, line 437: Because uveitis and retinitis are intraocular inflammatory conditions, please render “external disease, uveitis and retinitis” as “intraocular disease, uveitis and retinitis.

Page 12, line 496 - 499: Poxvirus Protein MC132 is one of the proteins; however, the authors should elaborate on these other proteins they are referring to in this sentence.

Page 14, line 599: Render “(vIFNa/BBP)” as “(vIFNα/βBP)”

MINOR ISSUES

Page 2, figure 1: Consider rendering “poor” as “low income”

Page 2, lines 64 - 68: The authors should reword this sentence to render it more comprehensible.

Page 4, line 173: DAI is not the appropriate abbreviation for dsRNA-activated (DAI) protein kinase. The appropriate rendering for DAI is DNA-dependent activator of interferon regulatory factor.

Page 7, figure 2: label the individual boxes and re-organize the current figure legend to reflect each content of the box.

Page 8 – 9, figure 3: The figure legend should be re-written to reflect the four subfigures. Provide a legend for 3i, 3ii, 3iii, and 3iv.

Page 8, line 310: Render “poly(IC)” as “poly(I:C)” The authors should ensure that all abbreviations used in the manuscript for the first time are written in full.

Page 9, line 330: Render “poly(IC)” as “poly(I:C)” The authors should ensure that all abbreviations used in the manuscript for the first time are written in full.

Page 9, lines 348 -350: Please verify the accuracy of this sentence.

Page 9, lines 376 – 377: Render “neutrophil granulocytes” as “neutrophils”

Page 10, line 414: Render “Jezek Z et al. (1988) showed” as “Jezek Z et al. showed”

Comments on the Quality of English Language

Page 2, lines 64 - 68: The authors should reword this sentence to render it more comprehensible.
